# SIGMA-GEN: STRUCTURE AND IDENTITY GUIDED MULTI-SUBJECT ASSEMBLY FOR IMAGE GENERATION

**Oindrila Saha**[*]
University of Massachusetts Amherst
osaha@umass.edu

**Vojtech Krs**
Adobe Research
vkrs@adobe.com

**Radomir Mech**
Adobe Research
rmech@adobe.com

**Subhransu Maji**
University of Massachusetts Amherst
smaji@umass.edu

**Kevin Blackburn-Matzen**[†]
Adobe Research
matzen@adobe.com

**Matheus Gadelha**[†]
Adobe Research
gadelha@adobe.com

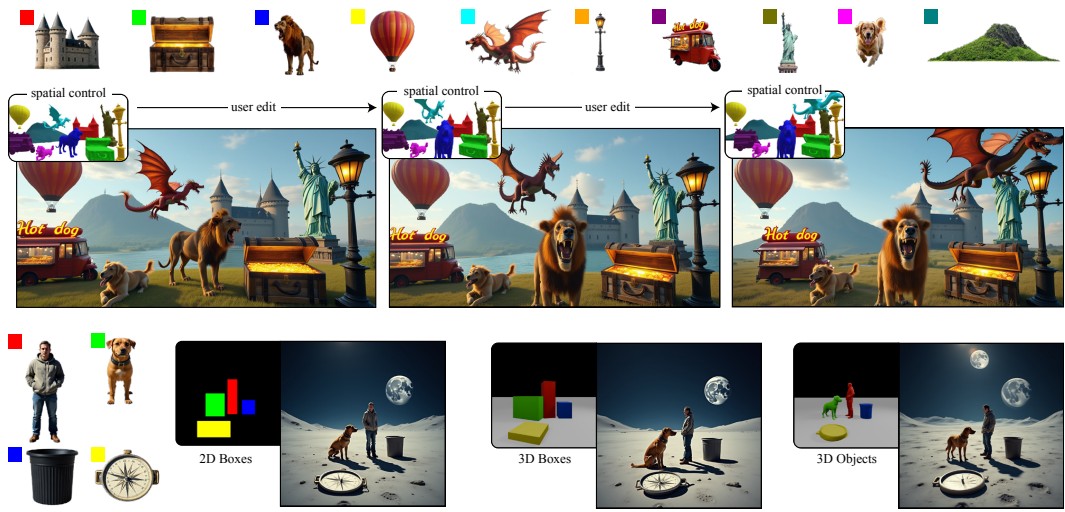

Figure 1: **SIGMA-GEN enhances controllability of text-to-image workflows by allowing users to prescribe both structure and subject identity.** In the top row, RGB images are used to describe subject identities. A 3D scene can be arranged by the user to describe the image structure; in these examples, meshes were automatically created using image-to-3D. The user can then assign identities to each subject (colors representing the assignments) and generate images while precisely editing the 3D scene. In the bottom part of the figure, we show that SIGMA-GEN can also be applied to simpler modes of structure guidance — 2D and 3D bounding boxes.

## ABSTRACT

We present SIGMA-GEN, a unified framework for multi-identity preserving image generation. Unlike prior approaches, SIGMA-GEN is the first to enable single-pass multi-subject identity-preserved generation guided by both structural and spatial constraints. A key strength of our method is its ability to support user guidance at various levels of precision — from coarse 2D or 3D boxes to pixel-level segmentations and depth — with a single model. To enable this, we introduce SIGMA-SET27K, a novel synthetic dataset that provides identity, structure, and spatial information for over 100k unique subjects across 27k images. Through extensive evaluation we demonstrate that SIGMA-GEN achieves state-of-the-art performance in identity preservation, image generation quality, and speed. Code and visualizations at https://oindrilasaha.github.io/SIGMA-Gen/.

---

[*]Work done during internship at Adobe Research
[†]Equal advising

# 1 INTRODUCTION

Recent advances in text-to-image generative models have enabled high-quality and diverse image synthesis from natural language prompts (OpenAI, 2023b; Rombach et al., 2022; Peebles & Xie, 2023; Lipman et al., 2022). However, these models still lack fine-grained control, which limits their adoption in real-world creative workflows. In particular, users have little ability to (i) control the identity of subjects and (ii) specify their arrangement within a scene.

We argue that both forms of control are essential. For identity, we introduce the use of single-view RGB images as descriptors, inspired by artistic workflows where exemplar images define visual elements for integration. For layout, we advocate for 3D object representations with rendered depth serving as a natural proxy for position, orientation, and occlusion relationships. To accommodate different levels of user expertise and control, we propose a single model that supports structural inputs at varying granularities: ranging from coarse 2D bounding boxes, to 2D masks, 3D bounding boxes, and per-pixel depth maps. This enables users to balance the ease of specification with the precision of control.

To the best of our knowledge, our work is the first to *jointly* support both structural guidance (capturing subject position, orientation, and occlusions) and identity guidance across multiple subjects within a single diffusion process. While prior works such as ControlNet (Zhang et al., 2023), Omni-Control v1/v2 (Tan et al., 2024), have demonstrated the use of multiple control modalities, none can reliably enforce multiple identities alongside structural layout guidance. Previous methods can be adapted to perform iteratively to address the issue of generating images with multiple subjects, but typically incur high runtime costs and suffer from compounding degradation in image quality. In contrast, our approach enables simultaneous generation of multiple identity-controlled subjects, arranged according to structural constraints, in a single forward pass.

The key to our approach is a large-scale synthetic data generation pipeline that automatically produces aligned RGB images, depth maps, masks, and identity descriptors. The pipeline leverages recent advances in image generation, grounding, and depth estimation. Using this pipeline, we construct a dataset containing **27k** images spanning over **100k** distinct identities, providing the supervisory signal required for training. We evaluate our models against competitive baselines and demonstrate significant improvements in both quality and controllability. When using per-pixel depth and masks, our method improves overall image fidelity by **31** points, identity preservation by **2** points, and achieves **4×** faster generation when synthesizing scenes with five or more distinct subjects. When analyzed under the coarser control modality of bounding boxes, our method improves overall image fidelity by **6** points and identity preservation by **11** points when synthesizing scenes with five or more distinct subjects. These results highlight the effectiveness of combining identity and structural guidance in a unified text-to-image generation framework.

# 2 RELATED WORK

**Structure control for image generation.** Photorealistic image generation with diffusion models has paved the way for integrating conditions beyond text (Ho et al., 2020; Rombach et al., 2022; Podell et al., 2023; Peebles & Xie, 2023; Lipman et al., 2022). ControlNet (Zhang et al., 2023) and T2I-Adapter (Mou et al., 2024) introduce auxiliary conditioning pathways that enable structural guidance from inputs such as edges, depth, or segmentation. Other diverse of control such as bounding boxes (Li et al., 2023b; Zheng et al., 2023; Wang et al., 2024c; Cheng et al., 2024; Chen et al., 2023), and 3D priors (Bhat et al., 2024; Omran et al., 2025; Ma et al., 2023) have also been explored. However, these methods do not provide mechanisms for controlling subject identity.

**Subject personalization.** DreamBooth (Ruiz et al., 2023) and Textual Inversion (Gal et al., 2022) showed that diffusion models can learn subject identity by training a map of a few images of a given subject to a unique text token. Subsequent methods (Kumari et al., 2023; Gu et al., 2023; Liu et al., 2023a;b; Zhu et al., 2025; Jang et al., 2024) extend personalization to multiple subjects within a single image by incorporating positional information in text, adapting weights, or encoding spatial layout. However, all these methods require per-subject optimization to learn their appearance.

**In-context generation.** Another line of work trains diffusion models to incorporate subject images as conditioning (Ye et al., 2023; Wang et al., 2024a; Wei et al., 2023). With the advent of

diffusion transformer (DiT) architectures, methods such as GPT1-Image (OpenAI, 2023a), Nano Banana (Google, 2025), and Flux Kontext (BFLabs et al., 2025) have been developed to support identity preservation. Recent open-source methods (Mou et al., 2025; Wu et al., 2025; Guo et al., 2025; Chen et al., 2025) extend identity preservation on diverse tasks controlled by text. However, these methods struggle to personalize across multiple subjects, as training datasets typically include only a small number of identities per image. Moreover, they lack support for structural control.

Subject insertion methods such as AnyDoor (Chen et al., 2024) and Insert Anything (Song et al., 2025) provide mask-guided control but are limited to inserting a single subject per generation. MS-Diffusion (Wang et al., 2024b) and other recent works (Tan et al., 2024; Tarrés et al., 2025; Li et al., 2023a; Xiao et al., 2025; Wang et al., 2025a) support different levels of spatial control, such as depth or bounding boxes. However, these methods are either limited to single-subject personalization or struggle with multi-subject personalization involving many identities. This can also be attributed in part to the scarcity of multi-subject training data. In contrast, SIGMA-GEN enables controllable multi-subject insertion, guided by identity images for each subject, within a single generation step (Figure 1). This leads to higher-quality and more coherent scene generation.

**Subject personalization datasets.** While several datasets support single-subject personalization–such as DreamBooth (Ruiz et al., 2023), CustomConcept (Kumari et al., 2023), AnyInsertion (Song et al., 2025), and Subjects200k (Tan et al., 2024)—only few focus on multiple subjects in an image. Virtual try-on datasets (Choi et al., 2021; Liu et al., 2016) contain 2–4 identities per image, including person identity and garments, but are limited to that domain. MultiWine (Tarrés et al., 2025) proposed a general-purpose dataset with up to two identities per image, obtained either from videos or through manual annotation. In contrast, we present the SIGMA-SET27K— a synthetically generated dataset with up to 10 subjects per image along with spatial controls and captions.

## 3 METHOD

Our objective is to generate an image $I$ that includes subjects $s_1, s_2, \ldots s_n \in \mathbf{S}$ based on a prompt $P$, and controls $C$, such that all subject identities are preserved and are placed according to spatial controls $C$. To enable controllable generation with many subjects in one shot, we create a first-of-its-kind, high-quality dataset (§ 3.1) containing multiple subject identities in each image. Furthermore, we propose a lightweight representation (§ 3.3) that enables multi-subject generation effectively and efficiently. Finally, our dataset and proposed representation is used for finetuning models with varying granularity of structural control (§ 3.4).

### 3.1 SIGMA-SET27K DATASET

We illustrate our pipeline for generating SIGMA-SET27K in Figure 2. For each target image, our dataset provides per-subject data including an identity image, mask, depth and 2D/3D bounding box. We begin by prompting an LLM to produce an image-generation prompt describing multiple subjects against diverse backgrounds, along with subject and background captions. For each prompt, we generate the target image using an off-the-shelf text-to-image model. Next, we use a grounded segmentation tool to generate individual subject masks using the subject captions. We also employ a depth estimation model to predict the target image depth. Next, in a key step of our pipeline, we repose the subjects using Flux-Kontext (BFLabs et al., 2025) to obtain identity images with varying poses and lighting conditions. Finally, to enable coarser control than per-pixel depth, we fit 2D and 3D bounding boxes to the segmented subjects. Please refer to the Appendix for more details.

### 3.2 THE SIGMA-GEN ARCHITECTURE

SIGMA-GEN takes as input a prompt $P$, a multi-subject identity control image $I^{\mathcal{S}}$, and a spatial control image $I^C$ to generate an image $I$. Both identity and spatial control images are encoded using the pre-existing VAE of a diffusion transformer model. Inspired by OminiControl (Tan et al., 2024), we adopt a unified attention mechanism in which the noisy image latents $X$ and conditions $P, I^{\mathcal{S}}, I^C$ are concatenated along the token dimension as $[P, X, I^{\mathcal{S}}, I^C]$, enabling multi-modal attention from every modality to all modalities.

Figure 2: **Pipeline for generating SIGMA-SET27K.** Our fully automatic synthetic data generation pipeline involves creating compositional prompts with an LLM, generating images from these prompts, segmenting to obtain subject crops, reposing the crops to produce identity images, and estimating depth and 3D bounding boxes. We also show an example of a training sample for fine control scenario of using precise masks and depth. The routing mask is colored to RGB for visualization purpose, the pixel values for the subjects being 10, 20, 30 in practice for this example.

## 3.3 REPRESENTING MULTI SUBJECT CONTROL

We decompose the spatial control $C$ into two categories: routing control $\mathcal{R}$ and structure control $T$. The routing control specifies where each subject should be placed in the image, while the structure control can define additional overall control for the scene such as depth. To efficiently represent the spatial controls $C \in [\mathcal{R}, T]$, we embed them into a single spatial control image $I^C$ with dimensions $H \times W \times 3$, matching the shape of the generated image $I$. We show an example of our control images in the right column of Figure 2, and in mask visualizations of Figure 3.

Let $\mathbf{S} = \{s_i\}_{i=1}^N$ denote the set of subjects and $\mathcal{R} = \{R_i\}_{i=1}^N$ denoting their desired spatial regions in the image domain. We define a mapping function $f : \mathbf{S} \longrightarrow \mathbb{M}$, which assigns each subject $s_i \in \mathbf{S}$ to a unique pixel intensity $m_i \in \mathbb{M}$. The control image $I^{\mathcal{R}}$ is constructed as $I^{\mathcal{R}}(x) = f(s_i)\mathbf{1}[x \in R_i]$. Thus, $I^{\mathcal{R}}$ with shape $H \times W$ encodes the subject layout, mapping each pixel to its corresponding intensity $m_i$, with 0 representing background. We discuss how we ensure each pixel maps to a single subject in the next section.

We now construct a subject identity condition image $I^{\mathcal{S}}$ (see training sample in right column of Figure 2). This image provides explicit identity information of each subject, allowing each pixel in the routing control to be unambiguously associated with its corresponding subject. Formally, let each subject $s_i$ be represented by an image $I^{s_i} \in \mathbb{R}^{H' \times W' \times 3}$ of fixed spatial resolution $H' \times W'$. We then define $I^{\mathcal{S}}$ by concatenating all subject images $\{I^{s_i}\}_{i=1}^N$ along the height dimension which yields $I^{\mathcal{S}} \in \mathbb{R}^{(N \cdot H') \times W' \times 3}$. Thus the $i$-th $H' \times W'$ block of $I^{\mathcal{S}}$ corresponds directly to region $\mathcal{R}_i$, serving as a compact visual dictionary of subject identities and regions.

## 3.4 MULTIMODAL STRUCTURE CONTROL

We design a method capable of handling diverse control modalities—including pixel-level masks or depth maps, 2D bounding boxes, and 3D bounding boxes—within the same condition $I^C$, by concatenating the routing control $I^{\mathcal{R}}$ and the structure control $I^T$.

The routing control image $I^{\mathcal{R}}$ can be created from pixel-wise region information $\mathcal{R}$. However, the effectiveness of this control image relies on the precision of the masks. Accurate masks naturally handle occlusions and ensure that no two regions overlap. In contrast, coarser forms of control such as 2D or 3D bounding boxes, may produce ambiguous overlaps. In crowded scenes, this often results in partial or complete occlusion. To address this limitation while preserving an efficient representation in which all spatial conditions are encoded within a single image, we adopt a bidirectional compositing strategy. Specifically, we construct two routing control images $I^{\mathcal{R}_{\mathrm{asc}}}$ and $I^{\mathcal{R}_{\mathrm{dsc}}}$ by pasting the subject masks in ascending order of subject occurrence in $I^{\mathcal{S}}$ and descending order respectively. The mask values $m_i$ associated with each subject $s_i$ remain constant across both constructions; only the order of composition is varied. This bidirectional compositing increases the chances of a region being visible in either of the two routing control images. The structure control $I^T$ refers to depth in our formulation and can be precise depths or coarser depths such as 3D bounding box depths. We refer the reader to Figure 4 for examples of coarser control images.

With the routing controls $I^{\mathcal{R}_{\text{asc}}}$ and $I^{\mathcal{R}_{\text{dsc}}}$ and structure control $I^T$ we can create the spatial control image $I^C = \text{Concat}(I^{\mathcal{R}_{\text{asc}}}, I^{\mathcal{R}_{\text{dsc}}}, I^T; \text{channel}) \in \mathbb{R}^{H \times W \times 3}$. For the case where we have precise masks of each subject, both the routing control images are the same $I^{\mathcal{R}_{\text{asc}}} = I^{\mathcal{R}_{\text{dsc}}}$. This representation leads us to having only two condition images – a *subject identity control image* $I^{\mathcal{S}}$ and a *spatial control image* $I^C$ which together can enable controllable multi-subject generation.

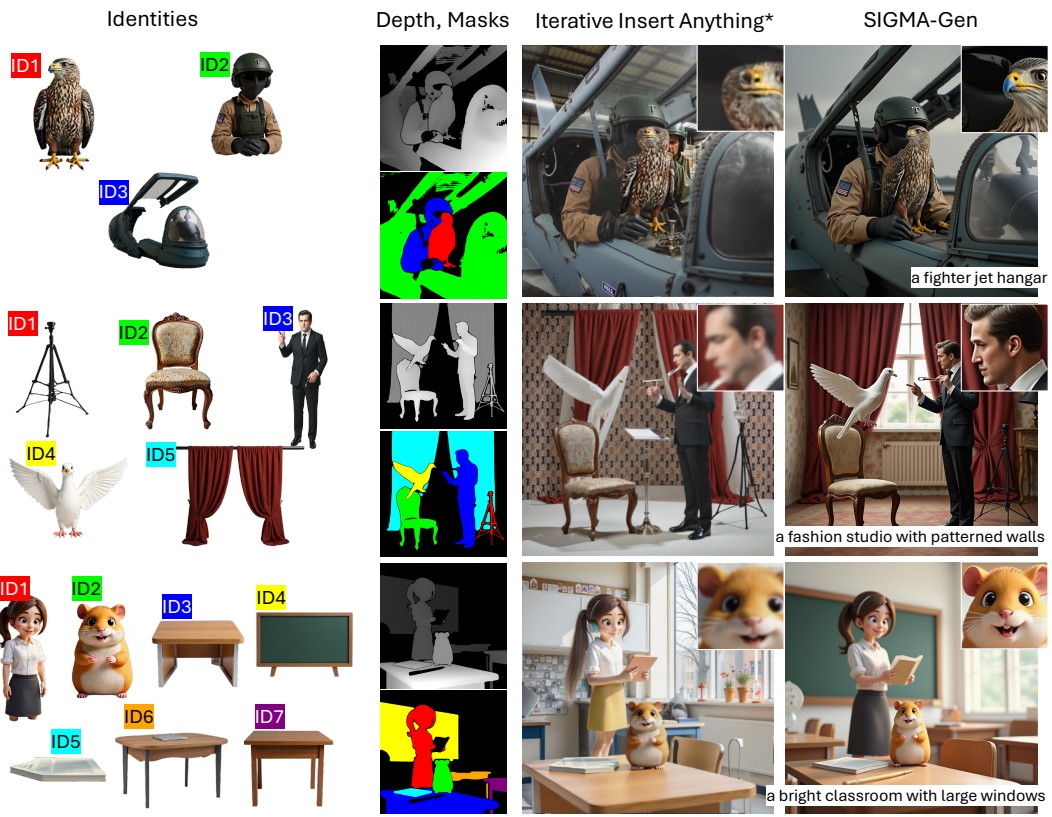

Figure 3: **Multi-subject generation with masks and depth.** SIGMA-GEN outperforms baselines both in terms of image quality (see zoomed crops at top-right) and subject identity preservation. For our case we prepend "Place these subjects in" to the prompts.

# 4 EXPERIMENTS

## 4.1 IMPLEMENTATION DETAILS

We adopt Flux.1 Kontext [dev] as the base model for training our method. For spatial control, we use the same RoPE (Su et al., 2023) embedding as that of the noisy image. For identity control, we also use the same RoPE but set the first dimension to ones (instead of zeros) to differentiate it from the noisy image, similar to Flux Kontext. Training is conducted on a single node with 8 A100 GPUs in three stages, using a LoRA rank and alpha of 128, the Prodigy optimizer (Mishchenko & Defazio, 2023), and a total batch size of 8. In the first stage, we train for 30k steps on a subset of data containing up to four subjects per image. The second stage trains for 20k steps on images with three or more subjects, followed by a third stage of 20k steps on images with more than four subjects.

To develop unified coarse-to-fine spatial conditioning, we randomly sample one of three structural inputs for each training example: (i) precise masks with depth, (ii) 3D bounding box masks with depth, or (iii) 2D bounding boxes. We further apply random dropping of one spatial condition channel with probability 0.1, as well as augmentations including random dilation of masks and bounding boxes and aspect ratio variation of bounding boxes by 1%, to improve robustness.

During training, we retain only the depths of the subjects, masking all other regions to zero. This mitigates the need to provide background depth at inference. For conditioning, we alternate between full prompts and background prompts with equal probability, constructing them as either *"Place these subjects in <bg prompt>"* or *"Place these subjects to compose: <full prompt>"*. For constructing the routing control image, we assign region intensity in steps of 10, with the region for the first subject marked with 10, the second subject with 20 and so on. We use non-gray colors while depicting masks in our figures for better visualization.

## 4.2 EVALUATION

For evaluation, we construct a dataset of 710 examples, including 200 single-subject personalized generation cases and 510 multi-subject cases. Dataset statistics are provided in the Appendix, and the data is generated as described in § 3.1.

We adopt DINO (Oquab et al., 2023) (DINO-I) and SigLIP (Zhai et al., 2023) (SigLIP-I) scores to evaluate subject identity preservation via image-to-image similarity. For multi-subject cases, we crop the generated image around the bounding box of each subject and compute similarity with the corresponding identity image, finally averaging over each crop and subsequently over every image. Cropping reduces the presence of other distracting elements in the image while computing similarity. We also evaluate text-to-image similarity using the SigLIP text-to-image score (SigLIP-T), which captures overall composition and background fidelity. We use SigLIP as it has been shown to outperform CLIP on fine-grained understanding tasks.

When precise depth is provided as control, we additionally compute the mean squared error (MSE) between the depth of the original and generated images, restricted to the subject regions. Unless otherwise specified, our evaluations use only subject depths as structure control. Finally, we assess perceptual quality with CLIP-IQA (Wang et al., 2023) and MUSIQ (Ke et al., 2021).

## 4.3 BASELINES

For single-subject personalization evaluation, we use OminiControl with its trained subject and depth LoRAs, as well as UniCombine. We also adapt a strong baseline, Flux Kontext, to incorporate depth guidance by attaching a depth ControlNet during inference. In addition, we design another strong baseline by combining the state-of-the-art mask-based insertion method, Insert Anything, with a depth ControlNet and a depth-to-image tool (BFLabs, 2024) to generate the initial image on which subjects are inserted. We refer to this variant as *Insert Anything\** and use it for both single- and multi-subject personalization.

For the multi-subject setting, iterative insertion is the only feasible baseline, as no prior work enables multi-subject personalized generation with precise masks and depth. We also adopt MSDiffusion as a baseline for the coarser control task of generation with bounding boxes, for both single- and multi-subject cases. In our setup, bounding boxes are provided as filled mask images (see § 3.4), pasted in ascending and descending order of subject occurrence in the identity image. This design choice enables a unified control modality, whereas MSDiffusion instead constructs box embeddings directly from bounding box coordinates.

## 5 RESULTS

We compare SIGMA-GEN with baselines both quantitatively and qualitatively. We further analyze different control types through ablations, showcasing emergent properties and applications.

## 5.1 COMPARISON WITH BASELINES

We present quantitative evaluation of our model with different types of controls in Table 1. Using a single trained model, we evaluate with (a) precise masks and depth, (b) 2D bounding boxes, and (c) 3D bounding box masks with depth. For this evaluation, SIGMA-GEN uses only subject depths, whereas other methods that accept depth are provided with full image depths. As shown in Table 3, SIGMA-GEN with subject depths performs comparably to the setting when full depth is provided. For the precise mask and depth case, we use background prompts for fair comparison

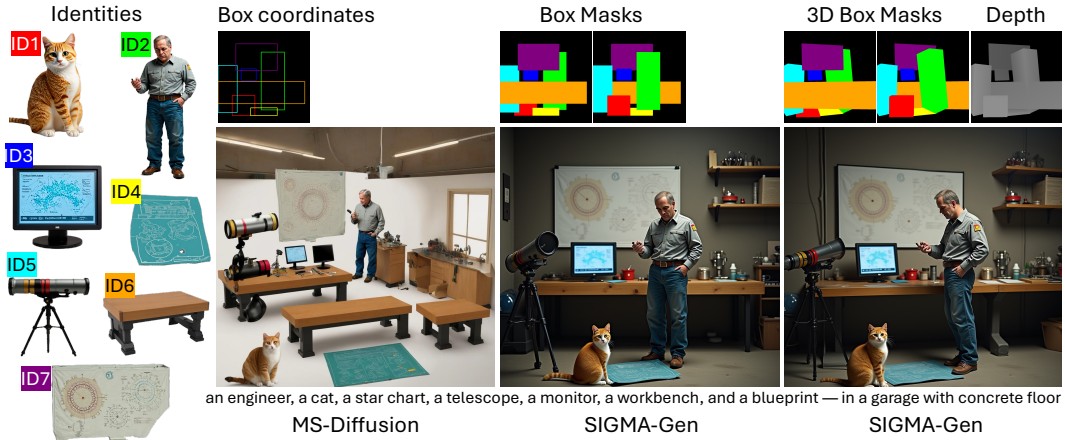

Figure 4: **Multi-subject generation with coarse controls.** Baseline fails to maintain position or identity, while SIGMA-GEN adheres to both 2D and 3D bounding-box coarse control. For our case we prepend "Place these subjects to compose: " to the prompt.

| | Method | Subject | | Text | Control | Depth | Quality | |
| | | DINO-I | SigLIP-I | SigLIP-T | | MSE | CLIP-IQA | MUSIQ |
|---|---|---|---|---|---|---|---|---|
| **Single Subject** | UniCombine | 75.89 | 82.72 | 15.52 | Depth | 103.1 | **71.47** | 70.81 |
| | OminiControl | 70.78 | 81.06 | 15.63 | Depth | 163.9 | 67.28 | 69.91 |
| | Insert Anything* | 74.69 | 81.44 | **16.01** | Mask, Depth | 216.2 | 65.97 | 67.35 |
| | Flux Kontext* | 79.96 | 83.22 | 15.99 | Depth | 112.4 | 66.30 | 64.93 |
| | SIGMA-GEN | **81.04** | **83.99** | 15.91 | Mask, Depth | **70.2** | 70.65 | **70.87** |
| | MSDiffusion | 74.83 | 82.65 | 2.41 | Bbox | - | 61.98 | 67.40 |
| | SIGMA-GEN | **79.16** | **83.70** | **16.13** | Bbox | - | **70.82** | **70.75** |
| | SIGMA-GEN | 79.98 | 84.00 | 16.03 | Mask, 3D bbox | - | 70.94 | 70.67 |
| **Multi Subject** | Insert Anything* | 72.72 | 75.58 | 17.66 | Mask, Depth | 203.4 | 44.41 | 48.86 |
| | SIGMA-GEN | **74.54** | **77.82** | **17.73** | Mask, Depth | **26.35** | **72.64** | **73.21** |
| | MSDiffusion | 63.28 | 69.06 | 11.20 | Bbox | - | 61.99 | 69.05 |
| | SIGMA-GEN | **71.90** | **73.15** | **17.21** | Bbox | - | **68.83** | **70.96** |
| | SIGMA-GEN | 73.48 | 75.27 | 18.19 | Mask, 3D bbox | - | 72.45 | 72.55 |

Table 1: **Quantitative comparison with baselines.** SIGMA-GEN achieves competitive or superior performance in single-subject controllable identity-preserving generation, and significantly outperforms baselines in the multi-subject setting.

to the baselines, while for the coarser controls we use full prompts as the bounding box baseline MSDiffusion expects full prompts too.

Our method consistently surpasses all baselines in subject consistency, as reflected by DINO-I and SigLIP-I scores for both single- and multi-subject personalization. For image-to-text similarity (SigLIP-T), our method outperforms all baselines except *Insert Anything*\* in the single-subject scenario. The stronger SigLIP-T performance of *Insert Anything*\* can be attributed to its use of a depth-to-image model with full depth to generate an initial image for subject insertion. Our ablation with full depth (Table 3) confirms that SigLIP-T benefits from access to full depth during generation.

Our method also achieves the lowest MSE between original and generated depths, demonstrating strong adherence to precise depth control while also supporting coarse depth guidance. In terms of image quality, measured by CLIP-IQA and MUSIQ, our approach is consistently competitive and often surpasses baselines across all control types. Notably, SIGMA-GEN performs strikingly well in the multi-subject setting, significantly outperforming all baselines across every evaluation metric.

Figure 5: **Performance over increasing number of subjects.** Baseline *Iterative Insert Anything\** run-time increases and quality decreases steeply compared to SIGMA-GEN. Our method also outperforms consistently in subject consistency and depth MSE.

| Controls | Subject | | Text | Depth | Quality | |
|---|---|---|---|---|---|---|
| | DINO-I | SigLIP-I | SigLIP-T | MSE | CLIP-IQA | MUSIQ |
| Mask (BG) | 74.17 | 77.52 | 17.62 | 40.10 | 71.26 | 72.27 |
| Mask + depth (BG) | 74.54 | 77.82 | 17.73 | 26.35 | 72.64 | 73.21 |
| Mask + depth (FULL) | 74.82 | 77.99 | 18.26 | 25.17 | 73.36 | 73.53 |

| Type of Depth Control | Subject | | Text | Depth | Quality | |
|---|---|---|---|---|---|---|
| | DINO-I | SigLIP-I | SigLIP-T | MSE | CLIP-IQA | MUSIQ |
| Subject depths (tokens) | 74.54 | 77.82 | 17.73 | 26.35 | 72.64 | 73.21 |
| Full depth (tokens) | 74.32 | 77.46 | 18.08 | 24.43 | 72.83 | 73.34 |
| Full depth (ControlNet) | 74.10 | 76.38 | 17.56 | 24.42 | 72.79 | 73.31 |

Table 2: **Ablation over increasing guidance.** (BG) represents text prompts describing only the background, whereas (FULL) describes the whole scene. Removing depth reduces performance while providing full prompts that include subject names improves performance.

Table 3: **Ablation over depth control.** Providing depth beyond the subjects to our method leads to better quality, depth alignment, and text alignment. Providing the full depth using a ControlNet leads to loss in subject consistency, and text alignment.

For multi-subject personalization with precise masks and depth, we plot SIGMA-GEN's performance as the number of subjects increases (Figure 5), comparing against the strong baseline of *Iterative Insert Anything\**. As expected, inference runtime grows much more steeply for the baseline due to repeated insertions. This iterative process also leads to a reduction in image quality, as reflected by lower CLIP-IQA and MUSIQ scores. In contrast, SIGMA-GEN maintains quality regardless of the number of subjects.

In terms of subject consistency, measured by DINO-I and SigLIP-I, SIGMA-GEN experiences some degradation as the number of subjects increases—largely due to dataset distribution effects (see Appendix)—yet still significantly outperforms the baseline. Text-to-image consistency scores remain similar between *Iterative Insert Anything\** and SIGMA-GEN, with our method achieving higher performance on average. Finally, our approach preserves spatial consistency across subjects, as shown by stable MSE values, while the baseline's performance degrades with more subjects.

We show qualitative comparison for multi-subject personalization with precise masks and depth in Figure 3. We show examples with three, five and seven subject insertion. For the baseline, with three and five subjects, we see a loss of identity in the eagle and dove respectively. For seven subjects, along with loss of identity of the person we also observe that some subjects do not get inserted such as the chalkboard and the last two tables. In contrast, SIGMA-GEN successfully follows all spatial and subject controls.

We qualitatively compare the coarser tasks of 2D and 3D bounding box control against the baseline MSDiffusion, which also relies on bounding boxes. We show a case of seven subject insertion in Figure 4. MSDiffusion fails to maintain correct position or identity of the subjects. In contrast, SIGMA-GEN follows spatial and identity controls for both types of control granularities. We show quantitative evaluation on DreamBooth Ruiz et al. (2023) dataset, human evaluation, comparison with commercial models, and additional qualitative results in the Appendix.

## 5.2 ABLATIONS

In Table 2 we probe the performance of our method for the precise mask and depth scenario by 1) removing depth entirely while maintaining precise masks and 2) by providing full scene description as input prompt. Note than in Table 1 we used background prompts for the precise mask and depth case (also second row of Table 2) for fair comparison to baselines, but here we use full prompts for ablation in the last row. We observe that including the full prompt improves scores across all

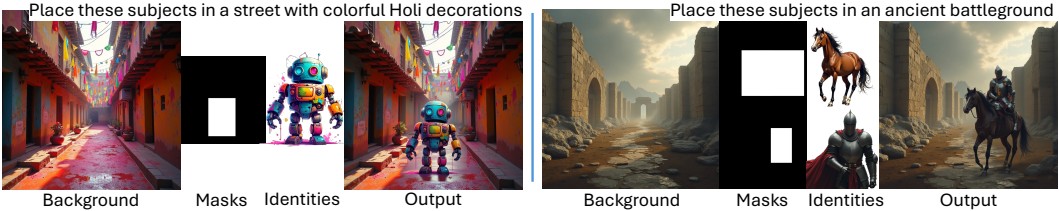

Figure 6: **Insertion using SIGMA-GEN.** We show extendability to one-shot single and multiple identity insertion given a background image based on bounding boxes.

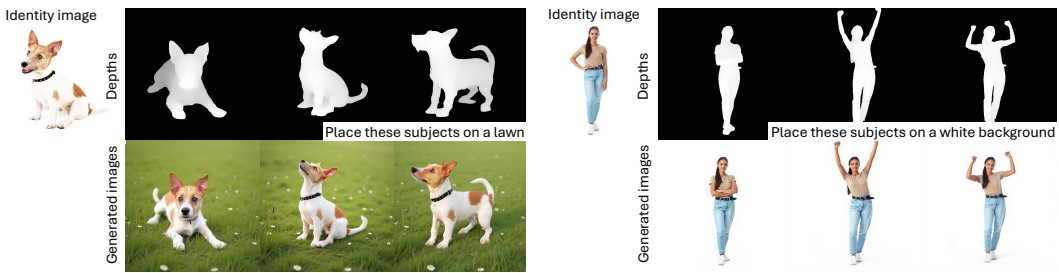

Figure 7: **Reposing subjects.** Through various poses supplied using depth, SIGMA-GEN can repose deformable subjects on different backgrounds.

dimensions compared to using only the background prompt. We also observe that removing depth but providing precise masks (first row) reduces performance as less information is provided.

Secondly, in Table 3 we probe various methods of providing depth control to our trained model. We observe that providing the full depth improves performance in all aspects except for subject consistency where it decreases slightly. However, all scores consistently surpass the baselines. We perform an experiment where we omit depth by passing the structure control image $I^T$ as all zeros, but pass depth externally through a pre-trained ControlNet. We observe degradation in subject consistency, text alignment and quality compared to providing the full depth via our structure control. We include additional ablations in the Appendix.

## 5.3 OTHER APPLICATIONS

**Insertion.** Although SIGMA-GEN has been trained for personalized image generation based on spatial/identity controls and prompt, we show that we can enable single and multi-subject one-shot insertion in Figure 6. For this task, we utilize blended diffusion (Avrahami et al., 2022) as a plug-and-play method to preserve the background. During inference, we follow the same strategy as we do for personalized generation, except we use the noised latents of the reference background image to preserve it . We show single and multi-subject insertion, where we observe that our method is able to harmonize the style of the subject to that of the background while maintaining correct shadows and lighting. It should be noted that the multi-subject insertion we show here is not iterative and is achieved in a single denoising loop.

**Reposing.** In Figure 7 we show control over a pose of an entity while maintaining its identity using depths of different poses. We show examples of deformable subjects whose identities across poses is more challenging to capture.

We show more emergent properties of SIGMA-GEN in the Appendix including the capability to handle free-form masks, style change, and multiple granularity levels of control in same generation. Additionally, we include qualitative examples on using real references, high number of subjects.

## 6 CONCLUSION

We present SIGMA-GEN, a unified framework for controllable multi-subject, identity-preserving image generation. Our single model can follow spatial controls across fine-to-coarse granularities while preserving both the identities and arrangements of multiple subjects. To support this, we introduce a large-scale synthetic data generation pipeline that produces SIGMA-SET27K, a dataset with up to 10 subjects per image, which we use to train SIGMA-GEN. Through extensive evaluation, we demonstrate that SIGMA-GEN consistently outperforms baselines, with especially strong gains in scenarios involving five or more subjects. Finally, we showcase additional applications of SIGMA-GEN, like subject insertion and reposing.

## 7 ACKNOWLEDGEMENT

SM and OS were supported in part by the National Science Foundation under Grant #2329927.

## 8 REPRODUCIBILITY

We provide details of our method for synthetic data generation in § 3.1 and in the Appendix. We also state the implementation details of our method trained on said dataset in § 4.1. Our model and code are open-sourced at `https://oindrilasaha.github.io/SIGMA-Gen/`.

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
