## A  APPENDIX

### A.1  ADDITIONAL DETAILS OF SIGMA-SET27K

We use Qwen-3-8B (Yang et al., 2025) for generating the full, background and subject captions. We prompt the LLM to generate image generation prompts with 3 to 10 subjects. Based on the individual subject captions, we use Grounded-Segment-Anything (Ren et al., 2024) to obtain segmentations of the subjects and masks. We apply multiple filtering criteria in this step. We remove any boxes that are less than 1% of the total image area and those that are greater than 40% of the image area. We also remove any duplicate or overlapping masks and only keep one of them. Finally, we only keep samples that have greater than 2 subjects per image. We use MoGe-2 (Wang et al., 2025b) for depth estimation. We estimated oriented bounded boxes using Open3D (Zhou et al., 2018) on depth for each subject. We repose each subject image using Flux.1-Kontext-dev to obtain identity images. Finally, we get 26435 images with a total of 105756 unique identities. We show the plot of our training data distribution in Fig. 8. For single and double subject data we process previously available datasets AnyInsert (Song et al., 2025) and MUSAR-Gen (Guo et al., 2025) to obtain spatial conditions and captions which we use only for the first stage of training, while using our dataset for the next two stages. For evaluation, since there exists no other dataset for multi-subject insertion, we generate the test set in a similar manner as described above. Our test set contains a total of 710 images and 2102 unique identities. We have 200 images containing one subject and we show the plot of the multi subject eval data in Fig. 8.

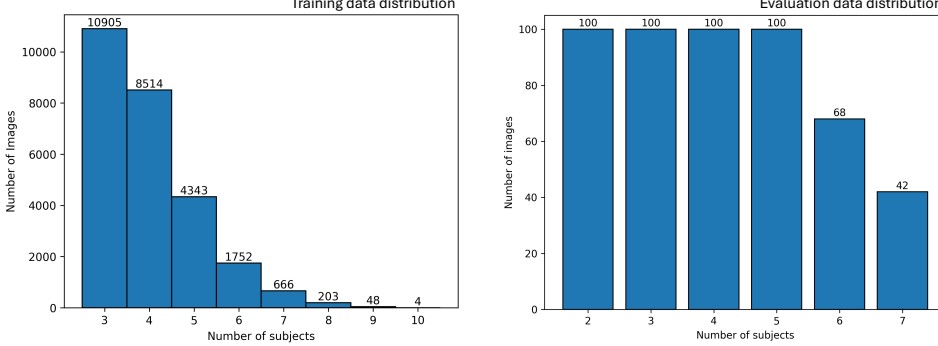

Figure 8: Train and test data distribution of number of images vs number of subjects they contain.

### A.2  ADDITIONAL ANALYSIS

In Figure 9 we visualize an attention map of a region of the generated subject, marked in the depth map and the generated image. We find the attention between the noisy image and the identity image and visualize the attention mean over the chosen pixel areas for the full identity image. We observe that the attention map has higher activation near the pink tongue area of the bag that we marked on the depth/generated image. Identity was sampled from DreamBooth (Ruiz et al., 2023).

We also investigated how providing depth information for areas outside the the subject masks affects the results. Even though SIGMA-GEN has been trained only for subject depths whose masks are also provided, it has learned to decouple the depth from the masks. We show this in Figure 10 where we progressively add regions to the depth while maintaining the same mask and a single identity image. We observe that SIGMA-GEN follows the provided depth strictly in all cases.

### A.3  ADAPTABILITY TO FREE-FORM MASKS DURING INFERENCE.

We probe the capability of our method to mask shapes unseen during training, such as circles or hand-drawn masks. We see in Fig. 11 that SIGMA-GEN can adapt to these forms of masks while maintaining identity and positions of subjects. Identities were sampled from DreamBooth.

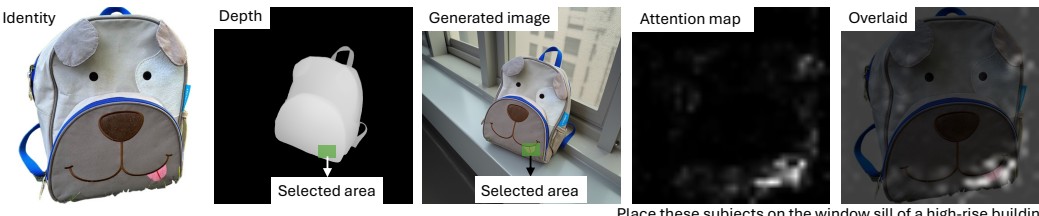

Figure 9: **Visualizing correspondence.** We visualize the attention map of a region marked in the noisy image space over the identity image.

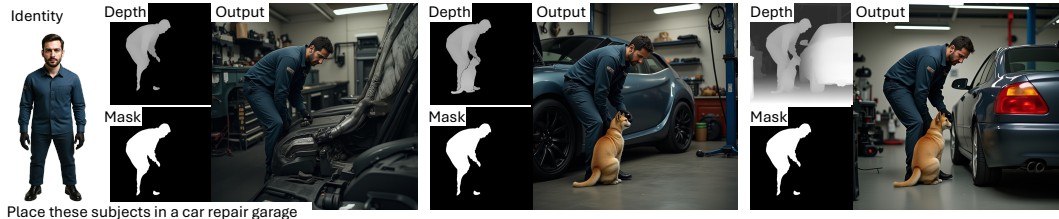

Figure 10: **Mask and depth are decoupled.** Despite being trained only with subject depths in the routing mask controls, SIGMA-GEN adapts effectively to increasing areas of depth control.

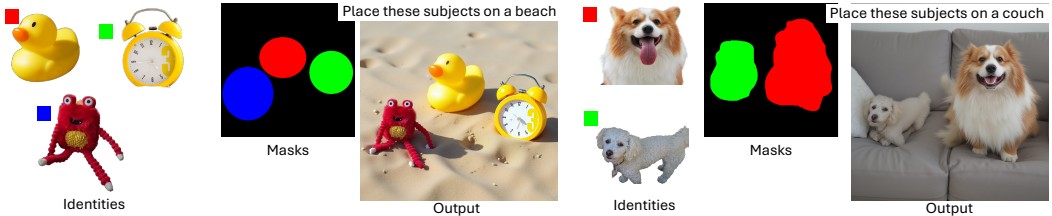

Figure 11: SIGMA-GEN can support circular or free-form masks during test time

## A.4 STYLE CHANGE OF SUBJECT THROUGH PROMPT

Fig. 12 illustrates style changes, where prompts such such as "Claymation" and "pencil drawing" alter the style of the subjects while maintaining overall identity.

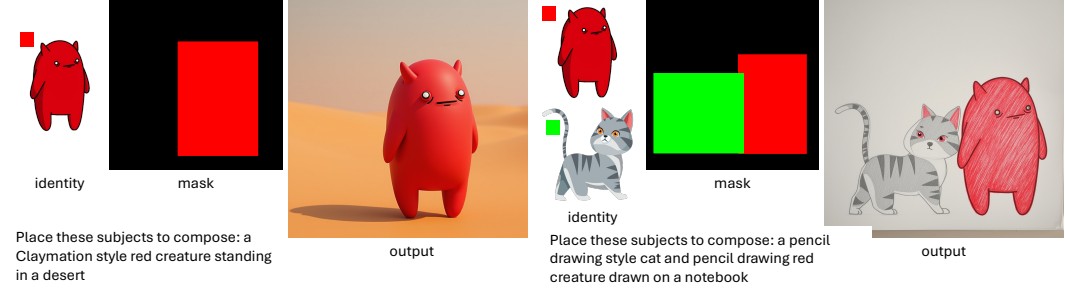

Figure 12: SIGMA-GEN can enable style change of subject(s) based on text.

## A.5 COARSE TO FINE CONTROLS IN SAME GENERATION

In Fig. 13 we show that we can provide 2D boxes, 3D boxes, precise mask, and pixel-level depth at the same time for different subjects in the same image generation loop. This allows flexibility to choose the control level per subject. Identities were sampled from DreamBooth.

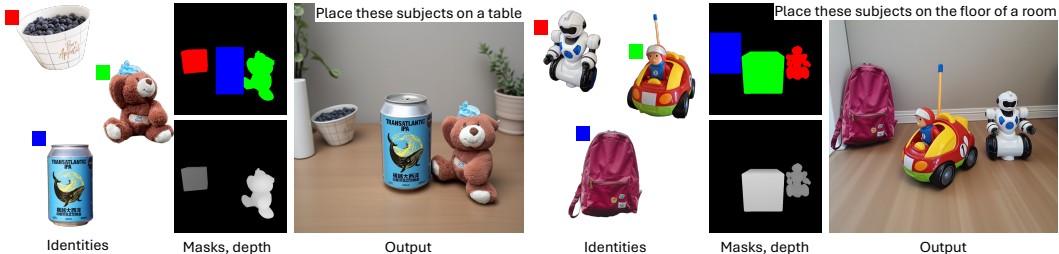

Figure 13: SIGMA-GEN can handle different types of conditions – pixel level mask, depth, 2D box, 3D box mask and depth – all at the same time.

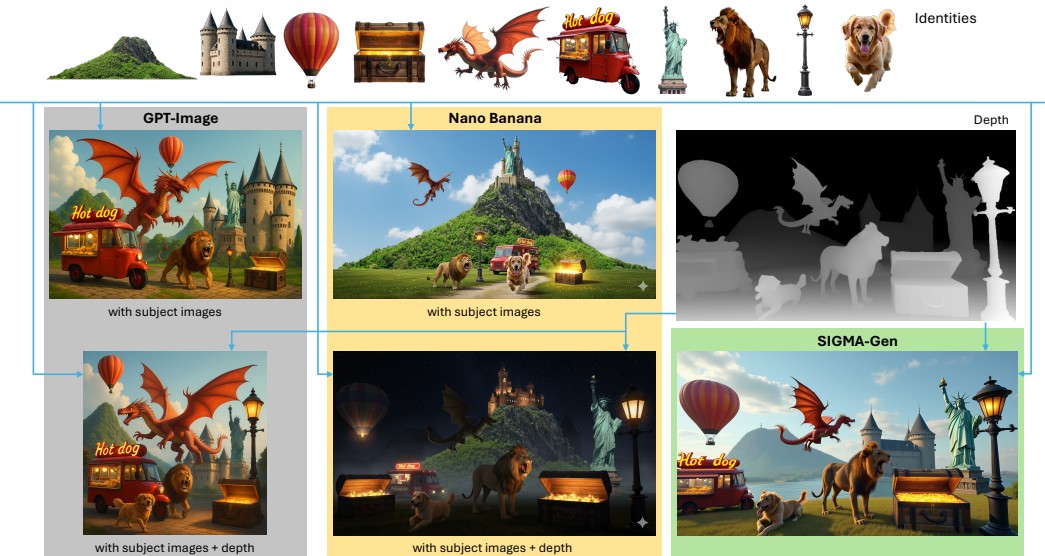

Figure 14: For 10 subject identity-preserved generation case, we test GPT-Image and Nano Banana.

### A.6    10 SUBJECT IDENTITY-PRESERVED GENERATION

For the 10 subject generation scenario of Fig. 1, we use Nano Banana (Google, 2025) and GPT-Image (OpenAI, 2023a). GPT-Image generates an unnatural looking image, does not follow depth, and also leaves out a subject (dog) in the top image. Nano Banana generates a high quality image and can follow depth to some extent, however we see identity loss especially in the hot-dog cart, hot air balloon, and castle.

### A.7    LLM USAGE

We use LLMs to find relevant prior work that could not be found with traditional search engines, to fix grammar and wording. We also used LLMs to brainstorm possible acronyms for the title. All outputs of the LLM were thoroughly verified by the authors before being added to the manuscript.

### A.8    FAILURE MODES

In Figure 15, we show a couple of failure modes of our method. Firstly, for coarser controls if the overlap among regions is too high e.g. in the case of the cyan and yellow areas, the model may ignore one of the subjects. Secondly, if the viewpoint of the subject to be generated is significantly different from that of the identity image, the subject consistency may lower, e.g. observe the paws of the lion which seem to be facing the front. We also observe loss in human facial identity as the training data is not specifically designed for this task.

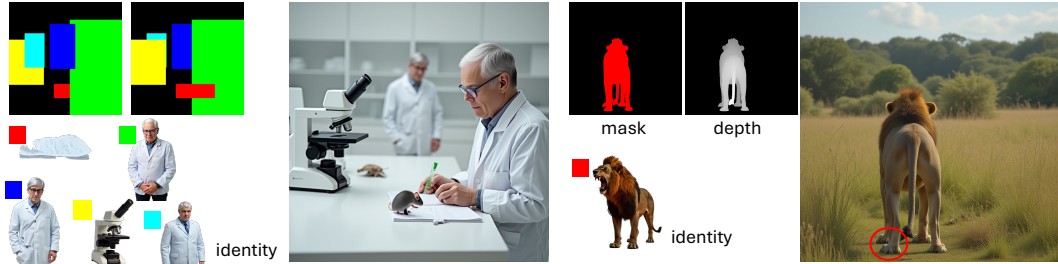

Figure 15: We show two failure cases of our method where subjects may be ignored in coarse control cases when the overlap is very high, and loss of subject consistency on high viewpoint change. On the example to the left, the significant overlap of the cyan box (scientist) and the yellow box (microscope) leads to the scientist not being generated. On the example to the right, even though the model generates a reasonable image with a viewpoint drastically different from the identity guidance, a closer look on the paws of the lion reveal they are actually pointing to the wrong directions (circled in red).

## A.9 MORE COMPARISON TO BASELINES

We plot evaluation metrics over increasing number of subjects for box control scenario and compare with MSDiffusion in Fig. 16. We see that subject consistency degrades for MSDiffusion more steeply compared to ours as subjects increase as evidenced especially by DINO score. SIGMA-GEN also surpasses MSDiffusion across text-alignment and quality.

We show qualitative comparison with baselines for the single subject controllable identity-preserved generation case in Fig. 17 for precise mask and depth and in Fig. 18 for coarse controls – 2D and 3D box. SIGMA-GEN consistently preserves identity, generates high quality images and follows control unlike baselines.

We show more comparison with baseline for multi-subject identity-preserved generation for coarse controls in Fig. 19. Again we observe that the baseline loses out on identity, quality, and correct positioning, often ignoring some subjects. For SIGMA-GEN we see that the 3D box generations get more positioning information in terms of relative depths of subjects compared to the 2D box scenario and correctly positions all the subjects following the depth.

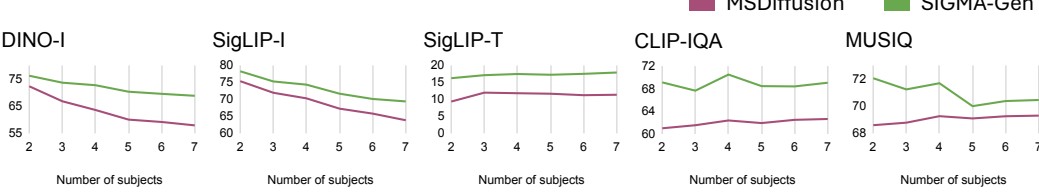

Figure 16: Comparison with baseline on coarse control multi-subject identity-preserved generation

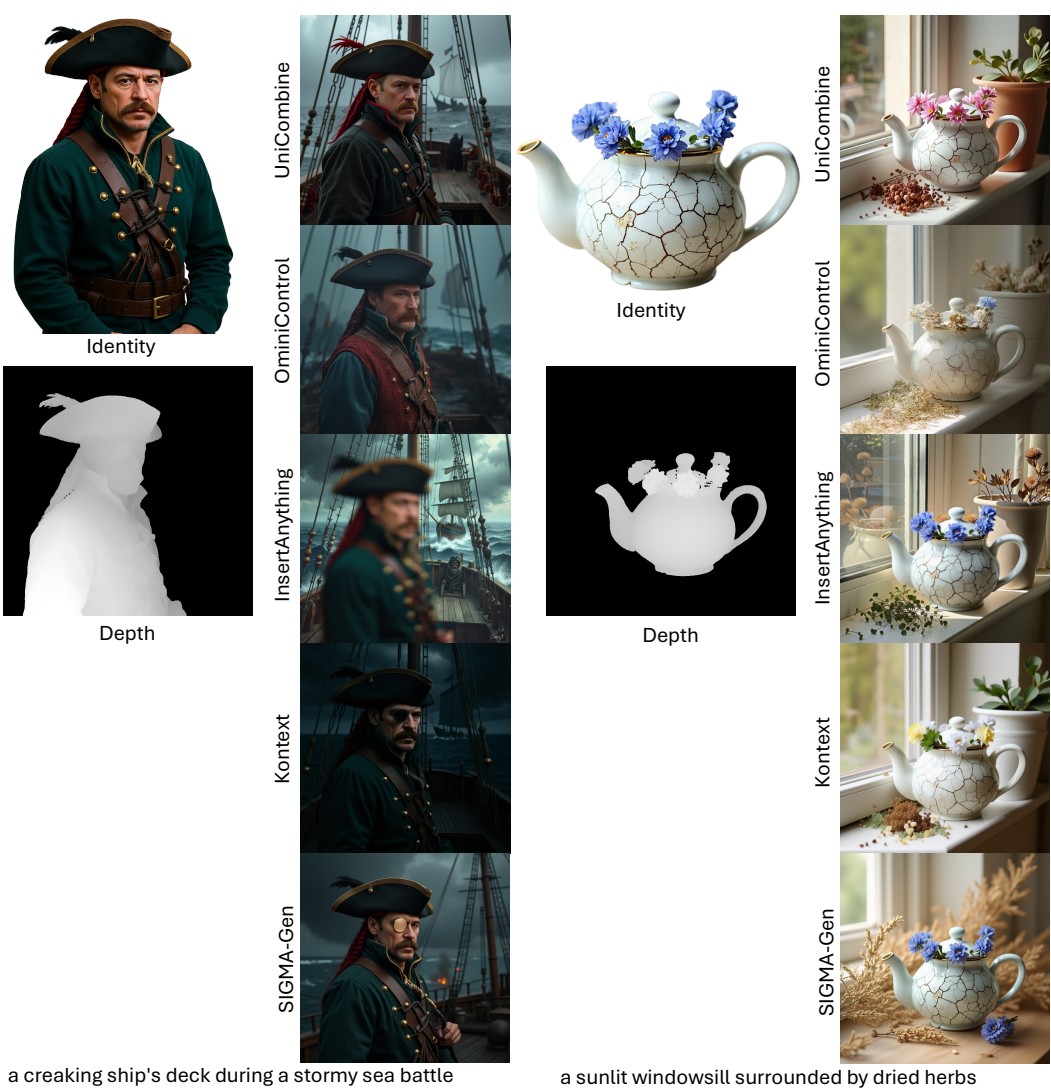

Figure 17: Qualitative comparison with baselines for single-subject generation in the case of precise mask and depth control. We prepend "Place these subjects in" to the prompts for our method.

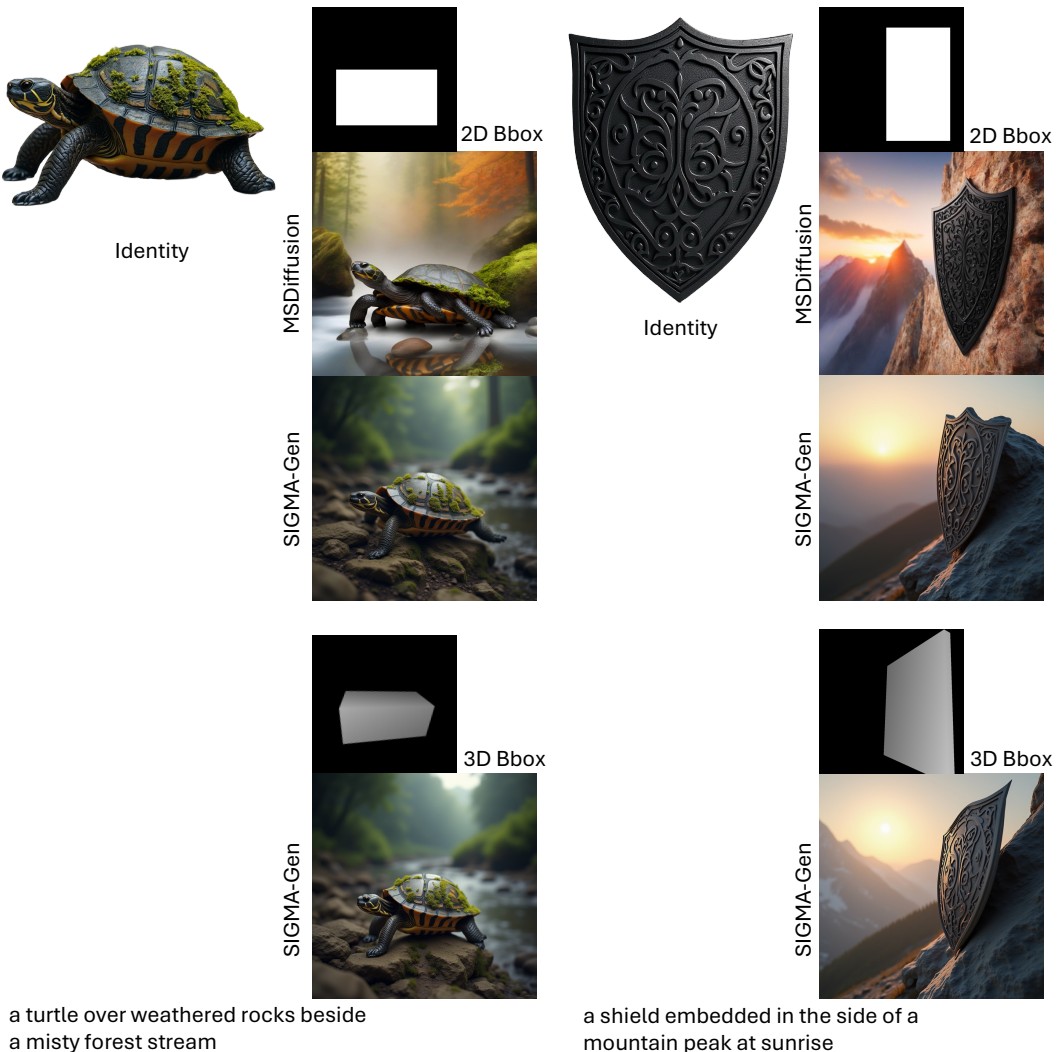

Figure 18: Qualitative comparison with baselines for single-subject generation in the case of coarse controls. We prepend "Place these subjects to compose: " to the prompts for our method.

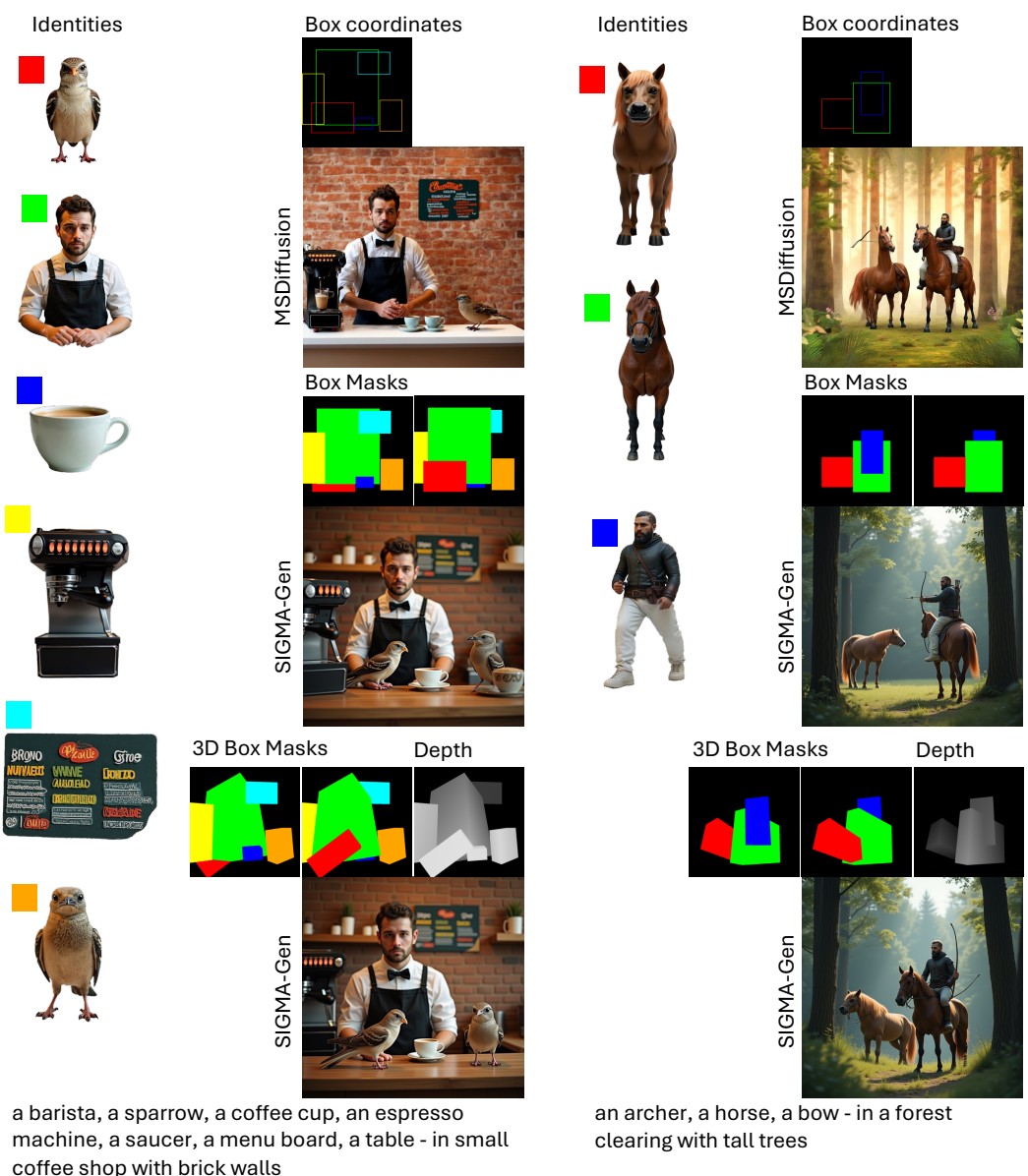

a barista, a sparrow, a coffee cup, an espresso machine, a saucer, a menu board, a table - in small coffee shop with brick walls

an archer, a horse, a bow - in a forest clearing with tall trees

Figure 19: More qualitative comparison results with baseline for multi-subject generation in the case of coarse controls. We prepend "Place these subjects to compose: " to the prompts for our method.

We conduct quantitative evaluation on the DreamBooth dataset Ruiz et al. (2023), where specifically we use the first image of every identity as the reference image and extract controls - depth and masks from all the other images which we treat as the targets. Similar to our data construction strategy described in Section A.1, we use MoGe-2 for depth estimation and Grounded-Segment-Anything for obtaining masks from both reference and target images. We also caption each target image using Qwen-2.5-VL. We use the same evaluation metrics as we did for our test datasets. In Table 4, we show that our method surpasses the previous baselines on the DreamBooth dataset as well.

| | Method | Subject | | Text | Control | Depth | Quality | |
|---|---|---|---|---|---|---|---|---|
| | | DINO-I | SigLIP-I | SigLIP-T | | MSE | CLIP-IQA | MUSIQ |
| Single Subject | Insert Anything* | 77.95 | 81.60 | 16.22 | Mask, Depth | 201.7 | 67.18 | 67.77 |
| | Flux Kontext* | 82.29 | 82.91 | 16.26 | Depth | 99.8 | 67.05 | 64.99 |
| | SIGMA-GEN | **82.84** | **84.06** | **16.27** | Mask, Depth | **58.4** | **71.43** | **70.32** |
| | MSDiffusion | 76.44 | 82.63 | 2.07 | Bbox | - | 62.79 | 66.42 |
| | SIGMA-GEN | **82.25** | **83.19** | **16.03** | Bbox | - | **71.50** | **70.19** |
| | SIGMA-GEN | 82.30 | 83.88 | 16.17 | Mask, 3D bbox | - | 71.62 | 70.21 |

Table 4: **Evaluation on DreamBooth dataset for single subject generation.**

We include additional qualitative results on using real references in Figure 20. For this figure, we sample identities from the Dreambooth and DeepFashion (Liu et al., 2016) datasets. Even though SIGMA-GEN has not been trained on datasets for human facial identity preservation, it handles person identities robustly. Performance on facial identity can be improved further with focused datasets in this domain. As stated in Section A.1, our training dataset also contains real domain single subject insertion - AnyInsert dataset samples which aids in robustness to real world samples.

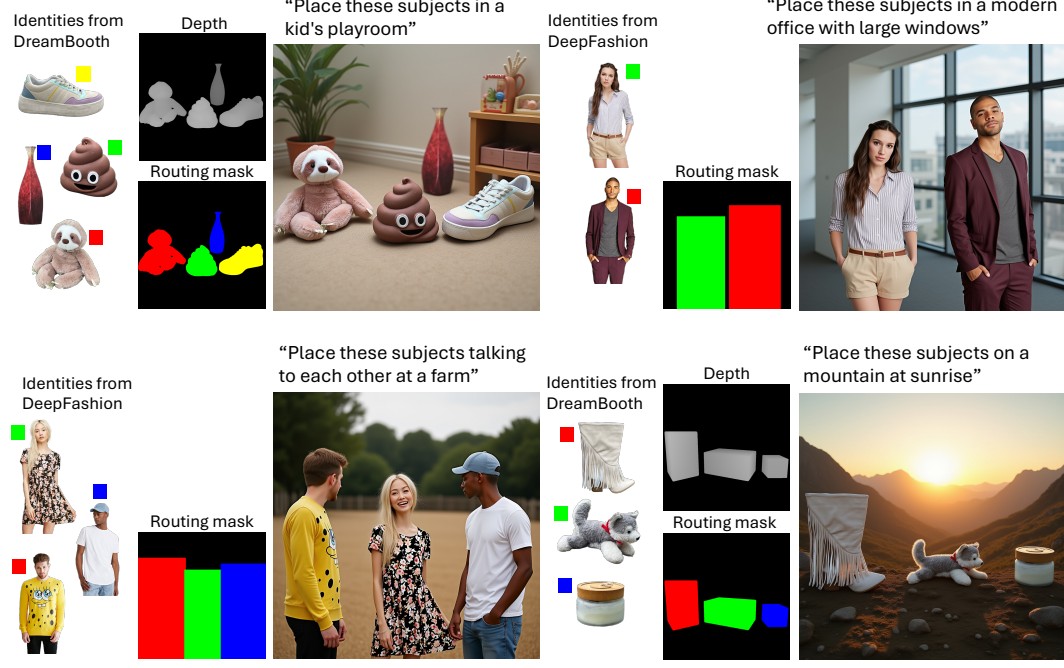

Figure 20: **Qualitative results with real reference images.** SIGMA-GEN robustly preserves identities of multiple real references across pose and viewpoint changes.

## A.12 ADDITIONAL QUALITATIVE COMPARISON WITH COMMERCIAL MODELS

In Figure 14 where we showed comparison to NanoBanana and ChatGPT for 10 subject controllable generation case. In Figure 21 we show that even with fewer number of subjects - 4 or 5 - both NanoBana and ChatGPT cannot adhere to the depth and also tend to lose identity (sloth toy in NanoBanana, vase in ChatGPT, red monster toy in NanoBana, IPA can in ChatGPT).

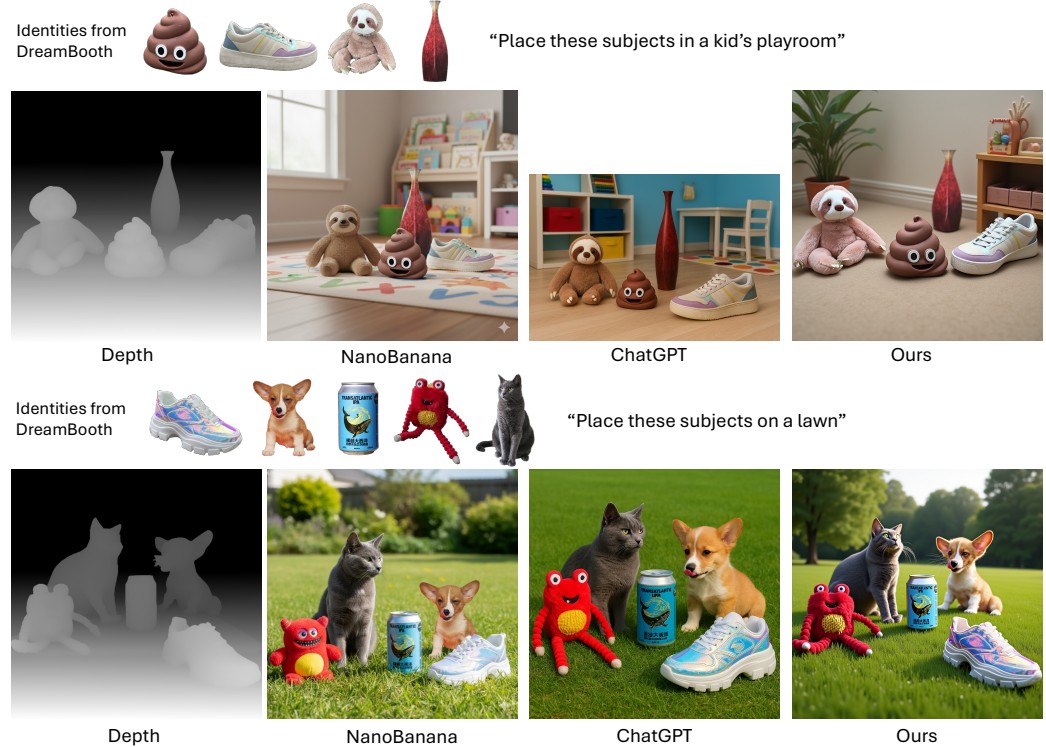

Figure 21: **Comparison to commercial models.** Unlike, NanoBanana and ChatGPT, SIGMA-GEN adheres to structure control and preserves identity of every subject.

## A.13 COMPARISON WITH INSERT-ANYTHING TRAINED ON OUR DATASET

We fine-tuned Insert Anything starting with their given model on SIGMA-SET27K. We conduct this training by choosing one identity from our images at a time randomly to train for insertion using the same strategy as Insert Anything for 5k steps (as suggested in their paper). We show the results of this experiment in Table 5 (row 2) for multi-subject insertion. We see slight improvement in identity preservation, and more improvement in image quality. However, this model still significantly lags behind SIGMA-GEN mainly owing to the iterative strategy of insertion for multiple subjects.

| Method | Subject | | Text | Depth | Quality | |
| | DINO-I | SigLIP-I | SigLIP-T | MSE | CLIP-IQA | MUSIQ |
| --- | --- | --- | --- | --- | --- | --- |
| Insert Anything* | 72.72 | 75.58 | 17.66 | 203.4 | 44.41 | 48.86 |
| Insert Anything$^{TR}$ | 72.84 | 75.65 | 17.64 | 187.3 | 49.93 | 52.06 |
| SIGMA-GEN | 74.54 | 77.82 | 17.73 | 26.4 | 72.64 | 73.21 |

Table 5: Effect of fine-tuning Insert Anything on SIGMA-SET27K

## A.14 ABLATION OVER ROUTING CONTROL

We explore different strategies for providing the routing control and show our ablation in Table 6. For the experiment with removing routing control (row 1), we omit the color assignment strategy and assign each box the same color intensity = 10. In row 2, we use the same order for pasting colored boxes for both channels (see Section 3.4). We observe that removing routing leads to a sharp decline in identity preservation. Removing biderection compositing also reduces identity preservation but less prominently, but image quality and text alignment remains similar. This can be associated to the model being able to generate most identities without bidirectional compositing but in erroneous locations and sizes.

| Method | Subject | | Text | Quality | |
| --- | --- | --- | --- | --- | --- |
| | DINO-I | SigLIP-I | SigLIP-T | CLIP-IQA | MUSIQ |
| without routing | 64.07 | 64.69 | 17.11 | 68.11 | 69.43 |
| without bidirectional compositing | 69.67 | 71.92 | 17.19 | 68.82 | 70.42 |
| proposed routing | 71.90 | 73.15 | 17.21 | 68.83 | 70.96 |

Table 6: Effect of routing strategy on multi-subject generation with 2D bounding boxes

## A.15 ABLATION OVER EFFECT OF CURRICULUM LEARNING

We show in Table 7 that on training with the complete dataset together reduces performance across all aspects for multi-subject generation using 2D bounding boxes. Since we start training with Flux Kontext which can handle a single reference image, we find it beneficial to gradually increase number of references in terms of reducing routing confusion and ensuring all subjects get included.

| Method | Subject | | Text | Quality | |
| --- | --- | --- | --- | --- | --- |
| | DINO-I | SigLIP-I | SigLIP-T | CLIP-IQA | MUSIQ |
| without curriculum learning | 65.36 | 66.39 | 17.04 | 67.94 | 68.47 |
| with curriculum learning | 71.90 | 73.15 | 17.21 | 68.83 | 70.96 |

Table 7: Effect of curriculum learning on multi-subject generation with 2D bounding boxes

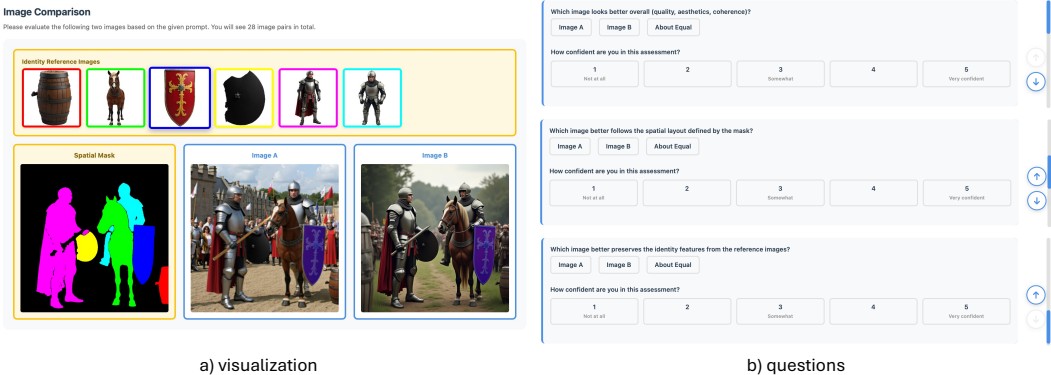

a) visualization      b) questions

Figure 22: **Setup of human study.** We illustrate our designed human study setup. a) We show identities, routing mask and generated images of Insert-Anything* and SIGMA-GEN. On hovering over an identity (shield selected in this case) the corresponding region in the generated images is highlighted for the users to compare. b) We show the questions we ask the users to evaluate the methods.

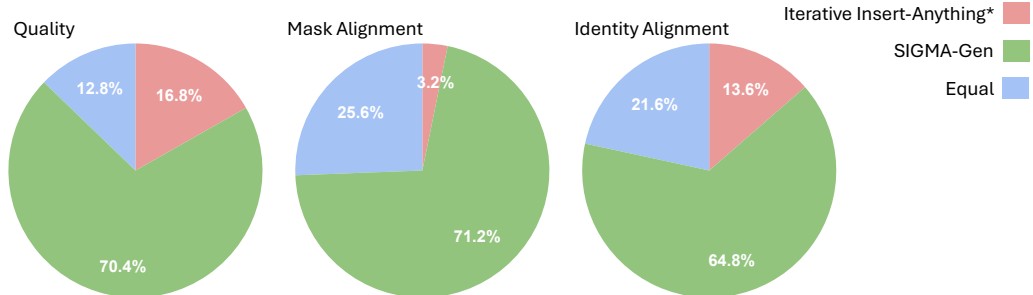

Figure 23: **Results of user study.** We show that users prefer SIGMA-GEN over the baseline more frequently in terms of image quality, alignment to mask control, and alignment to identity control.

## A.16    HUMAN STUDY

We show the setup of our designed human study in Figure 22. We designed this evaluation for using precise mask and depth with SIGMA-GEN and the baseline Insert-Anything* for multi-subject identity preserved generation. Our user study contains 30 samples of multi-subject generation - 5 each for two, three, four, five, six, and seven subject cases. We have obtained a total of 125 responses across 19 participants. We show the results in Figure 23. We observe that users preferred SIGMA-GEN in all three areas considered - quality, mask alignment and identity preservation.