# OpenReview forum: "SIGMA-Gen: Structure and Identity Guided Multi-Subject Assembly for Image Generation"
_ICLR.cc/2026/Conference — ICLR 2026 Poster_

### Official Review · Reviewer_i8jB · 2025-10-14

**Soundness:** 4
**Presentation:** 4
**Contribution:** 4
**Rating:** 6
**Confidence:** 4

**Summary:**

This paper presents SIGMA-GEN, a unified framework for multi-subject, identity-preserving image generation under flexible spatial and structural controls. The method allows users to specify both (a) subject identities using exemplar RGB images, and (b) structural arrangements through controls of varying granularity—from 2D bounding boxes to pixel-level depth maps and 3D layouts.

**Strengths:**

1. The paper elegantly bridges subject personalization and structural control, previously treated as separate problems.

2. The routing–structure control representation is compact and adaptable to multiple input modalities (2D/3D/depth).

3. The authors benchmark across single- and multi-subject settings, include runtime analyses, and evaluate both fidelity and identity preservation.

4. SIGMA-SET27K is systematically generated with aligned modalities, providing a valuable resource for future research.

5. The framework aligns well with creative workflows (scene layout, compositing, virtual try-on), making it broadly relevant beyond academic interest.

**Weaknesses:**

1. The synthetic dataset may not reflect real-world visual variability; generalization to real photos is untested.

2. All quantitative metrics are automated (DINO, SigLIP, MUSIQ); perceptual user studies would strengthen the claims.

3. Some baselines (e.g., Insert Anything*) are inference-level adaptations, not retrained for fairness—potentially favoring SIGMA-GEN.

4. The paper ablates over control granularity but not over architecture (e.g., without bidirectional compositing or without routing tokens).

**Questions:**

1. How does the model behave when trained on real datasets with authentic multi-person scenes (e.g., MS-COCO or Visual Genome)?

2. Can the routing–structure image representation generalize to video or temporal settings?

3. Are there failure cases when subject identities overlap or occlude one another heavily?

4. How does the method handle domain shift between synthetic and real identities?

5. Could the authors release data-generation scripts separately from trained models for transparency?

---

> ### Author Response · Authors · 2025-11-20
> **Official Rebuttal for Reviewer i8jB**
>
> We thank the reviewer for appreciating the contributions of our method and dataset design, and for identifying the relevance of our work in creative workflows. We appreciate the reviewer’s suggestions and answer their queries below:
>
> - **W.1. Evaluation on real references.** We would like to point to Figures 9,11,13 that we originally provided where we tested SIGMA-Gen for various scenarios using DreamBooth identities. We also included a new Figure 20 (Section A.11) where we include additional qualitative examples on DreamBooth and DeepFashion datasets’ real world reference images for multi-subject generation. While these highlighted some qualitative samples, we conduct evaluation on the DreamBooth dataset for single-subject generation and discuss results in Section A.10. Both the qualitative and quantitative results point to SIGMA-Gen’s robustness to real scenarios. We would also like to highlight that we use the real domain AnyInsertion dataset for single subject generation as a part of our training data as we stated in Section A.1.
>
> - **W.2. User study.** Thank you for the suggestion. We have designed and started conducting a user study for the precise masks and depth scenario to compare SIGMA-Gen with the baseline Insert-Anything*. We show our human study setup in Section A.16 (Figure 22). We will update the main pdf with results as soon as it is complete. [EDIT: We have updated our main pdf with the user study results]
>
> - **W.3. Unfair comparison to Insert Anything.** We fine-tune Insert-Anything with our SIGMA-Set27k dataset and discuss the methodology of training and results on multi-subject generation in Section A.13. We find that fine-tuning on our data improves the quality of the Insert-Anything baseline. It also leads to slight improvement in identity preservation. However, it still lags behind SIGMA-Gen significantly owing to the losses accumulated by repetitive insertion.
>
> - **W.4. Ablation over routing control.** In Table 6 we show that removing routing control decreases identity preservation significantly and also leads to loss in image quality. Removing the bidirectional compositing strategy also leads to loss in identity preservation performance. We discuss this in Section A.14.
>
> - **Q.1. Training on real data.** We are unable to train on MS-COCO or Visual Genome due to the lack of reference images for identities in the dataset. This refers to the fact that for every element in the image, no varied viewpoint or pose exists of the same identity that can be used for identity preserved training. However, we would like to point to Figure 20 which shows the robustness of SIGMA-Gen to real world reference images taken from DeepFashion dataset.
>
> - **Q.2. Extension to video generation.** Since DiT architectures for image and video generation share many of the same design principles, we expect our approach to generalize to videos as long as video-specific routing and structure are provided. The primary limitation is access to large-scale video data. While our data-generation strategy could, in principle, be extended to videos, we defer this exploration to future work.
>
> - **Q.3. Failure Modes.** In Figure 15 we showed a couple of failure cases where 1) on high occlusion the model can omit generating a subject, 2) very high viewpoint change may result in some inconsistencies.
>
> - **Q.4.  Robustness to real references.** The ability of SIGMA-Gen to work robustly in real domains could be partially attributed to the inclusion of the real world AnyInsertion dataset for single subject generation as a part of our training data as we stated in Section A.1.
>
> - **Q.5.  Data release.** We will release dataset generation, training, and inference code upon acceptance.
>
> Thank you for your valuable suggestions and we anticipate your thoughtful responses.

---

> > ### Comment · Reviewer_i8jB · 2025-11-25
> >
> > Thanks for your response. I will keep the rating.

---

### Official Review · Reviewer_UzUG · 2025-10-30

**Soundness:** 3
**Presentation:** 2
**Contribution:** 3
**Rating:** 6
**Confidence:** 4

**Summary:**

The authors introduce SIGMA-GEN, a unified framework designed for generating images containing multiple, specific identities. This method is presented as the first to achieve single-pass, multi-subject generation that preserves identity while adhering to both structural and spatial constraints provided by the user. A significant feature is the model's flexibility, allowing user guidance at various levels of precision (from simple bounding boxes to detailed segmentation maps) within one model. To accomplish this, the authors also created a large-scale synthetic dataset, SIGMA-SET27K, providing rich identity and spatial information. The paper claims state-of-the-art performance in identity preservation, image quality, and speed.

**Strengths:**

- The primary strength is its reported ability to handle multi-subject identity preservation in a single pass while simultaneously respecting complex structural and spatial constraints. This addresses a major limitation in existing generative models.
- The model offers significant practical utility by accepting a wide spectrum of user guidance—from coarse 2D/3D boxes to precise pixel-level maps—within a single, unified framework. This versatility makes it accessible for different use cases without needing specialized models.
- The creation of the SIGMA-SET27K dataset is a valuable contribution in its own right. A large-scale synthetic dataset with comprehensive annotations for identity, structure, and spatial information is a key enabler for this type of complex, multi-modal training and will likely benefit the wider research community.

**Weaknesses:**

- Lack of Clarity in Methodology and Reporting: The paper's presentation is not as clear as it could be. Specifically, the pipeline for constructing the SIGMA-SET27K dataset is underspecified; critical details about the tools and processing steps used are omitted, making the dataset's creation difficult to reproduce or fully evaluate.
- Insufficient Experimental Baselines: The experimental comparison, while showing strong results against some methods, is not comprehensive. To truly validate the "state-of-the-art" claim, the evaluation needs to be broadened to include more recent and relevant open-source methods. A comparative analysis against top-tier commercial models (such as those from Google, e.g., gemini-2.5-flash-image-preview, or JiMengAI) is also necessary to properly contextualize the model's performance.

**Questions:**

I am interested in the model's performance on more fine-grained, real-world customization tasks. Specifically, how effectively does this framework handle high-fidelity, instance-level customization, such as preserving the exact facial identity of a specific, real-world person provided by a user.

---

> ### Author Response · Authors · 2025-11-20
> **Official Rebuttal for Reviewer UzUG**
>
> We thank the reviewer for recognizing the novel and challenging problem area of multi-subject controllable image generation that we tackle, SIGMA-Gen’s ability to flexibly handle various granularity levels of control, and the contribution of our data generation pipeline. We address the mentioned weaknesses and questions below:
>
> - **W.1. Dataset generation details.** We would like to refer the reviewer to Section A.1 where we had provided details of the generation pipeline of SIGMA-Set27k. Additionally, we will be releasing data generation code upon acceptance.
>
> - **W.2. Comparison to baselines.** Since the area of structure and spatial controlled multi-subject generation is underexplored, we compared with all the recent and best prior methods available in this direction to the best of our knowledge. We request the reviewer to kindly point us to any relevant baselines we may have missed. We would also like to point to Figure 14 where we had provided comparison with ChatGPT and NanoBana. We have also included a new Figure 21 (Section A.12) where we include more qualitative examples. Since we are unable to access JimengAI, we provided comparison with ChatGPT and NanoBanana (gemini-2.5-flash-image-preview) and show that they suffer from identity inconsistency when handling multiple subjects, and also cannot adhere to depth maps strictly.
>
> - **Q.1. Evaluation on real references.** We refer the reviewer to the newly added Figure 20 which shows how SIGMA-Gen performs on multi-subject generation with identities sourced from the DeepFashion dataset of real people images. We observe that even though SIGMA-Gen has not been trained with focused people specific datasets, it can handle these cases robustly. Although it must be noted that performance on facial identity preservation can be significantly improved on training with human multi-view image datasets such as DeepFashion.
>
>
> Thank you for your valuable suggestions and we anticipate your thoughtful responses.

---

### Official Review · Reviewer_cKMR · 2025-10-30

**Soundness:** 3
**Presentation:** 2
**Contribution:** 2
**Rating:** 4
**Confidence:** 3

**Summary:**

This paper presents SIGMA-GEN for multi-subject identity-preserving image generation with structural control at various granularities. The authors introduce SIGMA-SET27K, a synthetic dataset with 27k images containing up to 10 subjects, and demonstrate improvements over baselines especially in multi-subject scenarios.

**Strengths:**

- Jointly controlling multiple subject identities and spatial layout at various granularities (2D boxes to pixel-level depth) with a single model addresses a practical need in creative workflows.

- Strong quantitative results with 31 points improvement in fidelity and 4x speedup for 5+ subjects, with consistent gains across metrics.

- Well-designed automatic data pipeline and comprehensive ablations provide good insights into component contributions.

- Demonstrated versatility through applications like insertion, reposing, and mixed-granularity control.

**Weaknesses:**

- Limited technical novelty since the core architecture directly adopts OminiControl's [1] unified attention mechanism without significant modification. The main contribution is dataset engineering rather than methodological innovation.

- Training and evaluating entirely on synthetic data is problematic. You're essentially fitting to outputs from Flux Kontext [2] and other models [3,4], then testing on the same distribution. This doesn't demonstrate real-world generalization, and you should validate on existing benchmarks like DreamBooth [5] with real photographs. Errors from the generation pipeline also propagate into your model.

- Critical technical details are underspecified. How does the model map routing mask intensities (10, 20, 30...) to identity image blocks - is this explicitly supervised or purely learned? The three-stage curriculum suggests fundamental scalability issues rather than unified learning. Why can't you train end-to-end?

- Incomplete baseline comparisons. You cite MultiBooth [6] and other recent multi-subject methods but only compare against MSDiffusion [7]. Also adapting MSDiffusion to use filled masks instead of coordinate embeddings may unfairly disadvantage it. No user studies validate whether your automatic metrics correlate with human perception.

- Training with only subject depths when Table 3 shows full depth improves results (SigLIP-T: 17.73→18.08) seems arbitrary. The bidirectional compositing for occlusions feels like a heuristic workaround rather than principled depth reasoning like InstanceDiffusion [8].

### References

[1] Tan et al., "OminiControl: Minimal and Universal Control for Diffusion Transformer," arXiv 2024

[2] BFLabs et al., "Flux.1 Kontext: Flow Matching for In-Context Image Generation," arXiv 2025

[3] Ren et al., "Grounded SAM: Assembling Open-World Models for Diverse Visual Tasks," arXiv 2024

[4] Wang et al., "MoGe-2: Accurate Monocular Geometry with Metric Scale," arXiv 2025

[5] Ruiz et al., "DreamBooth: Fine Tuning Text-to-Image Diffusion Models," CVPR 2023

[6] Zhu et al., "MultiBooth: Towards Generating All Your Concepts in an Image," AAAI 2025

[7] Wang et al., "MS-Diffusion: Multi-Subject Zero-Shot Image Personalization," arXiv 2024

[8] Wang et al., "InstanceDiffusion: Instance-Level Control for Image Generation," CVPR 2024

**Questions:**

- Can you evaluate on real-world data with actual photographs as identity images to validate generalization beyond the synthetic training distribution? How does performance compare when the identity images and test scenarios come from real captures rather than model-generated content?

- Please clarify the identity routing mechanism mathematically and show whether the three-stage curriculum is necessary with ablations. Why not compare to other recent multi-subject methods you cite like MultiBooth, and can you provide user studies validating your metrics?

---

> ### Author Response · Authors · 2025-11-20
> **Official Rebuttal for Reviewer cKMR**
>
> We thank the reviewer for appreciating SIGMA-Gen’s ability to support multiple granularity levels of control with a single model, its superior quantitative performance, versatile applicability, as well as our proposed data generation pipeline. We address the reviewer’s concerns below:
>
> - **W.1. Novelty.** While there is no architectural difference wrt OminiControl, the way control signals are passed as input is, to the best of our knowledge, novel -- while we recognize this is a very simple solution, it is also surprisingly effective. We hope our work can act as a strong baseline and inspire more sophisticated work tackling similar tasks in the future.
>
> - **W.2. Evaluation on real references.** Thank you for the suggestion. We have conducted evaluation on the DreamBooth dataset for single-subject generation and added in the results in Section A.10. We would also like to point to Figures 9,11,13 that we originally provided where we tested SIGMA-Gen for various scenarios using DreamBooth identities. We also included a new Figure 20 (Section A.11) where we include additional qualitative examples on DreamBooth and DeepFashion datasets’ real world reference images for multi-subject generation. We would also like to highlight that we use the real domain AnyInsertion dataset for single subject generation as a part of our training data as we stated in Section A.1.
>
>
> - **W.3.**
>
>   - a) **Clarification of routing control.** The ability to route identities based on the colored routing mask is purely learned by the model without any additional supervision. This can be attributed to the unified attention mechanism which allows the model to learn associations across both identity and routing conditions over image tokens. As described in detail in Section 3.3 our routing control is simply an image with masks of different intensities composited on it. This mask is then (optionally) concatenated with depth, encoded by VAE, and introduced into the unified attention mechanism. We show an example of this mask in Figure 2 where we use high contrast colors for each region, however as stated in its caption we use pixel values of 10, 20, 30, and so on for each region.
>
>   - b) **Curriculum learning.** Since we start fine-tuning over the Flux-Kontext model which can handle only one reference image, we find it beneficial to gradually increase the number of references. We provide an ablation as per your suggestion in Section A.15.
>
> - **W.4.**
>    - a) **Comparison to baselines.** We use prior in-context learning methods as baselines which can use a single image as reference at test-time. We do not use optimization baselines such as CustomDiffusion or MultiBooth due to the two directions being orthogonal. Since the area of structure and spatial controlled multi-subject generation with single reference images  is underexplored, we compared with all the recent and best prior methods available in this direction to the best of our knowledge. We request the reviewer to kindly point us to any relevant baselines we may have missed.
> We would also like to clarify that *we used MS-Diffusion in their default setting where they use bounding box coordinates* as we state in Section 4.3. We will make this point clearer in this section.
>
>    - b) **User Study.** We have designed and already started conducting a user study for the precise masks and depth scenario to compare SIGMA-Gen with the baseline Insert-Anything*. We show our human study setup in Section A.16 (Figure 22). We will update the results of the human study in the main pdf as soon as it is completed. [EDIT: We have updated our main pdf with the user study results]
>
> - **W.5.**
>    - a) **Training only subject depth is arbitrary.** We trained the model without providing background depth to improve the performance in real use cases where the practitioner only has structural data over the main subjects in the scene, but the background is to be fully generated from text guidance alone. With the popularization of text/image-to-3D models, this is a very common scenario – most of the models are object-centric, which makes subject meshes easier to obtain while scene-based counterparts are not as mature.
>
>   - b) **Ablation over bidirectional compositing.** Please refer to Section A.14. where we show the effectiveness of our bidirectional compositing strategy.
>
> Thank you for your valuable suggestions and we anticipate your thoughtful responses.

---

### Official Review · Reviewer_ETmR · 2025-10-31

**Soundness:** 3
**Presentation:** 2
**Contribution:** 2
**Rating:** 4
**Confidence:** 4

**Summary:**

The method focuses on the problem of fine-grained control in text-image generation. Existing methods support the function of controlling the single-subject structure or identity, but cannot handle the multi-subject structure and consistent identity with the accurate spatial layout. The proposed SIGMA-GEN proposed several modules to tackle the balance between accuracy and efficiency in multi-subject generation.

**Strengths:**

Strength:
-	The paper is well-motivated with several technical challenges;
-	The method proposed seems sound and correct.
-	The new evaluation sub-benchmark is proposed with a pipeline for the dataset generation.

**Weaknesses:**

Weakness:
-	It seems confusing that which exact module corresponds to tackle the problem of multi-subject structure and identity. It seems that the proposed modules are not unique to this specific challenge.
-	The novelty of the pipeline is limited. It seems integration of existing modules into a pipeline. Please directly compare with existing baseline methods and show the novelty of each module.

**Questions:**

See weakness

---

> ### Author Response · Authors · 2025-11-20
> **Official Rebuttal for Reviewer ETmR**
>
> As described in Section 3, the identity, routing and structure controls provided to the unified attention mechanism enable multi-subject controllable generation. The routing control is specifically designed to enable mapping different identities to the specific regions they should be generated in. Although our architecture matches OmniControl, our method of passing control signals is, to our knowledge, novel. Despite its simplicity, it proves surprisingly effective, and we hope it serves as a strong baseline that inspires more advanced approaches. Our proposed SIGMA-Set27k is also a novel contribution as no such dataset exists in literature which contains identity reference, structure, and position for up to 10 subjects per image. We evaluate our method with the state of the art in multi subject controllable image generation and would like to kindly request the reviewer to provide references to any relevant baselines that we may have missed. Thank you and we anticipate your thoughtful responses.

---

### Author Response · Authors · 2025-11-20
**Addressing reviewers' comments and invitation for discussion**

We would like to thank the reviewers for their valuable suggestions. The reviewers appreciated our novel data generation pipeline, the importance of the task being tackled, and the robustness and flexibility of SIGMA-Gen. We have now appended our previous supplementary material to the main pdf and also added new sections to the main pdf based on the suggestions by the reviewers. These new sections start from A.10. (page 21) and have the text “POST REVIEWS” prepended to the section names. We would like to address the most frequent concerns below:


- **Novelty.**
SIGMA-Gen is the first model that enables multi-subject identity preserved image generation while supporting diverse granularity levels of control. While our architectural changes are minimal, our novel and simple way of providing routing control to the model is highly effective. We also propose SIGMA-Set27k - a first of its kind dataset that includes up to 10 identities per image along with their references, structure and position information.

- **Performance on real references.**
We would like to point to Figures 9,11,13 that we originally provided where we tested SIGMA-Gen for various scenarios using DreamBooth identities. We also included a new Figure 20 (Section A.11) where we include additional qualitative examples on DreamBooth and DeepFashion datasets’ real world reference images for multi-subject generation. While these highlight qualitative aspects, we conduct evaluation on the DreamBooth dataset for single-subject generation and discuss results in Section A.10. Both the qualitative and quantitative results point to SIGMA-Gen’s robustness to real references. We would also like to reiterate that we use the real domain AnyInsertion dataset for single subject generation as a part of our training data as we had previously stated in Section A.1.


- **User study.**
We have designed and started conducting a human study for the precise masks and depth scenario to compare SIGMA-Gen with the baseline Insert-Anything*. We show our human study setup in Section A.16 (Figure 22). We will update the main pdf with the results of the study as soon as it is completed.  [EDIT: We have updated our main pdf with the user study results]

---

### Comment · Area_Chair_3M7S · 2025-11-24
**Discussion with Authors**

Dear Reviewers,

The authors have diligently provided responses to your questions and concerns. I request you to please review the authors' responses, acknowledge that you have read them and actively engage with them in further discussion as needed.

This discussion period, with the authors, will end on December 2, 2025 (AoE). However, I request that you not wait until the last minute and actively engage with the authors early.

Best,
AC

---

### Author Response · Authors · 2025-12-02
**Summary of our Rebuttal**

Dear Chairs,

We sincerely appreciate your dedication to the ICLR 2026 conference. For your convenience, we have prepared a concise summary of reviewers' comments along with our corresponding responses, as outlined below.

**Strengths**

The reviewers appreciated that our work **tackles a challenging and practically important problem by unifying multi-subject identity preservation with structural and spatial control within a single model**, capable of handling guidance ranging from coarse 2D/3D boxes to pixel-level depth maps (Reviewers ETmR, cKMR, UzUG, i8jB). They also highlighted that our **method is technically sound and achieves strong quantitative gains in fidelity and speed**, supported by thorough ablations and runtime analyses (Reviewers  ETmR, cKMR, i8jB). The reviewers also valued **the contribution of the SIGMA-SET27K dataset and its automatic, well-designed generation pipeline**, noting that it provides an aligned, multi-modal resource that enables diverse applications and benefits the broader research community (Reviewers ETmR, cKMR, UzUG, i8jB).

**Weaknesses**

Reviewers ETmR and cKMR noted concern over **novelty**. In our global official comment, we emphasize that **SIGMA-Gen and SIGMA-SET27K mark the first integrated effort in advancing multi-subject image generation with structural and spatial control**, establishing a foundation for future research in this area.

Reviewers cKMR, UzUG, i8jB requested **evaluation with real world reference images**. We also addressed this in our global official comment pointing to Figures 9,11,13,20 where we show **qualitative results on multi-subject generation** and Section A.10 where we showed **quantitative evaluation on the existing single-subject dataset DreamBooth**. It should be noted that **no multi-subject image generation datasets exist in prior art**.

Reviewers cKMR, i8jB suggested a **human study** to gauge preference of real users. We conducted a user study based on the suggestion and report the results in Section A.16.

Additionally, we clarified questions about the design of our routing control to Reviewer **cKMR** (W3.a), fair comparison to baselines (W4.a), and the significance of training with only subject depths (W5.a). We also provided **additional ablations** on the importance of curriculum learning (W3.b) and bidirectional compositing for routing mask construction (W5.b).

We pointed to the comprehensive details we had provided about the SIGMA-SET27K dataset in the supplementary for reviewer **UzUG** (W1). Along with previous qualitative **comparison with commercial models** - NanoBana and ChatGPT in Figure 14, we also added new examples in Figure 20 as requested by the reviewer (W2).

We **fine-tuned the previous baseline Insert-Anything on SIGMA-SET27K** as suggested by Reviewer i8jB (W3). We added an **additional ablation** stressing the importance of our routing control (W4). Additionally, we clarified the reviewer’s question on **training on real datasets** (Q1), which is not possible due to the **lack of such datasets which contain identity images for multiple subjects** in literature. We pointed to the fact that SIGMA-Gen works well for real identity images. We answered multiple queries regarding extension to video generation (Q2), failure modes (Q3), the reason for SIGMA-Gen’s robustness to real references (Q4), and that we will be releasing our dataset and models upon acceptance (Q5).

We have also updated our main pdf to incorporate reviewers’ suggestions. The post review material can be found in sections starting from A.10. (page 21) and have the text “POST REVIEWS” prepended to the section names. We thank you for your service.

Best,

Authors

---

### Meta-Review · Area_Chair_ZAGN · 2025-12-07

**Summary:**

This paper proposes SIGMA-GEN, a practical solution for multi-subject identity-preserving generation with flexible structure and spatial control. The main weakness is that parts of the contribution may still be viewed as incremental in method design and rely heavily on the synthetic data setting.

**Reviewer Concerns:**

The rebuttal basically resolves Reviewer i8jB’s concerns, while the other reviewers did not provide rebuttal-stage comments.

**Reviewer Scores:**

After considering the reviews and the rebuttal, I recommend acceptance as poster.

---

### Decision · Program_Chairs · 2026-01-26

Accept (Poster)